# NiCoCrFeY High Entropy Alloy Nanopowders and Their Soft Magnetic Properties

**DOI:** 10.3390/ma17020534

**Published:** 2024-01-22

**Authors:** Donghan Jiang, Zhifen Yuan, Zhenghou Zhu, Mengke Yao

**Affiliations:** School of Physics and Material Science, Nanchang University, Nanchang 330031, China; j14917646642022@163.com (D.J.); zhuzhenghou@ncu.edu.cn (Z.Z.); ymk18325837424@163.com (M.Y.)

**Keywords:** high entropy alloy, soft magnetic composites, soft magnetic properties

## Abstract

High entropy alloy nanopowders were successfully prepared by liquid-phase reduction methods and their applications were preliminarily discussed. The prepared high entropy alloy nanopowders consisted of FeNi alloy spherical powders and NiFeCoCrY alloy spherical powders with a particle size of about 100 nm. The powders have soft magnetic properties, the saturation magnetization field strength were up to 5000 Qe and the saturation magnetization strength Ms was about 17.3 emu/g. The powders have the excellent property of low high-frequency loss in the frequency range of 0.3–8.5 GHz. When the thickness of the powders coating was 5 mm, the powders showed excellent absorption performance in the Ku band; and when the thickness of the powders coating was 10 mm; the powders showed good wave-absorbing performance in the X band. The powders have good moulding, and the powders have large specific surface area, so that the magnetic powder core composites could be prepared under low pressure and without coating insulators, and the magnetic powder cores showed excellent frequency-constant magnetization and magnetic field-constant magnetization characteristics. In the frequency range of 1~100 KHz; the μ_m_ of the magnetic powder core heat-treated at 800 °C reached 359, the μ_e_ was about 4.6 and the change rate of μ_e_ with frequency was less than 1%, meanwhile; the magnetic powder core still maintains constant μ_e_ value under the action of the external magnetic field from 0 to 12,000 A/m. The high entropy alloy nanopowders have a broad application prospect in soft magnetic composites.

## 1. Introduction

In the past three decades, most of the work on magnetic composites has been focused on the development of powder cores, also known as soft magnetic composites (SMCs) [1]. SMCs have been made of micron-scale particles (usually Fe, but also MPP, Fe-P, Fe-Si, or Fe-Co), but only recently have researchers started to study magnetic composites using nanoparticles, which opened up a new direction for the introduction of nanoparticle-based magnetic composites in SMCs.

Based on the size of the magnetic particles, the composite core (micron or nanoscale) can be regarded as a distributed air gap inductor with an air gap of 10 to 10^8^ cells/cm. In addition to increasing the saturation magnetic field, the distributed air gap also promotes soft saturation. When near saturation, this condition causes a slow decrease in μr, thereby reducing the abrupt and destructive nature of saturation.

Through the transition to nanoparticle-based magnetic materials, there is an opportunity to further improve the properties of magnetic composites. However, there are very few public reports on this area. The transition from micrometer scale particles to nanoscale particles makes full use of decades of continuous research of isolated magnetic nanoparticles [2]. There are some differences between the composites made using micron-scale particles and nanoparticles. In some ways, nanocomposites are a continuation of the development trend of large-scale materials. For example, eddy current losses can be reduced to a negligible extent by reducing the particle size. However, the reduction of the particle size also introduces new physics principles [3,4].

Recently, the development of high entropy soft magnetic materials has been rapid [5,6]. High entropy alloys break through the limitations of traditional alloy components, and by adjusting the combination and content of multiple components, they are endowed with excellent mechanical and magnetic properties [7], such as high strength [8,9,10], high toughness [11,12,13], high hardness [14], corrosion resistance [15,16,17], and radiation resistance [18], and have shown great potential for application in many fields [19,20,21,22,23,24,25]. There are three main representative types of high entropy alloys: Cantor alloys (CoCrFeMnNi), which are dominated by 3d transition group metals; Senkov alloys (NbMoTaW), which are dominated by refractory metals; and low density, high entropy alloys (AlMgLiZnCu, AlMgZnCuSi, AlZrTiNbMo) [26,27], which are dominated by lightweight elements, such as aluminium, magnesium and titanium. Among them, Cantor alloys are based on three ferromagnetic elements, Fe, Co and Ni, and one or more non-ferromagnetic elements, such as antiferromagnetic Cr, antimagnetic Cu, Si, Ga, paramagnetic Al, Ti, Mn, Sn, etc., are added in trace amounts to improve the magnetic and mechanical properties of the alloys, so as to obtain high entropy soft magnetic materials with excellent comprehensive performance [13,28,29].

At present, high entropy alloys are prepared by methods such as atomization and mechanical alloying, and the particle size of the powder is greater than 50 μm. Report on the preparation of few high entropy alloy nanopowders [9,12,17,30,31,32,33,34]. NiCoCrFeY high entropy alloy nanopowders (herein after referred to as high entropy alloy nanopowders) are difficult to prepare and technically complex, so it is very important to explore a new simple and effective preparation method of practical and production significance [35,36].

In this paper, high entropy alloy nanopowders with the shape of nano-spherical powder was successfully prepared by liquid-phase reduction method, a new process for the preparation of high entropy alloy nanopowders was developed, and the soft magnetic properties of high entropy alloy nanopowders were preliminarily explored.

## 2. Experimental

### 2.1. Preparation and Characterization of High Entropy Alloy Nanopowders

Ni_40_Co_19.5_Cr_19.5_Fe_19.5_Y_1.5_ alloy nanopowders were prepared by the liquid-phase reduction method. Dissolved NiSO_4_•6H_2_O, FeSO_4_•7H_2_O, CrCl_3_•6H_2_O, CoCl_2_•6H_2_O, YCl_3_•6H_2_O in a certain proportion in water. After stirring at 80 °C, NaOH solution was added and reacted to produce Ni(OH)_2_, Fe(OH)_2_, Cr(OH)_3_, Co(OH)_2_, and Y(OH)_3_. Then, N_2_H_4_•H_2_O was added and the mixture was stirred for 40 min. A black suspension of FeNiCoCrY alloy nanopowders was obtained through static precipitation and cooling. After magnetic separation, the alloy nanopowders were cleaned multiple times with anhydrous ethanol.

The physical phase structure of the high entropus solution of NiSO_4_, FeSO_4_, CoCl_2_, CrCl_3_, and YCl_3_ was prepared according toy alloy nanopowders was analyzed by a Bruker-axe D8 advance X-ray diffraction analyzer (XRD, CuKα) with the following test conditions: tube voltage of 40 kV, current of 30 mA, and scanning speed of 0.02°. SEM, Scios2, and CZ type cold field emission scanning electron microscope were used to analyze the micro-morphology and energy spectrum of the high entropy alloy nanopowder. The thermogravimetric analysis of NiCoCrFeY high entropy alloy nanopowders was carried out using a German STA 449 F5 integrated thermal analyzer, with the samples protected by N_2_ at a flow rate of 100 mL/min, and with α-Al_2_O_3_ as the reference material, the temperature range was 30–800 °C, and the temperature increase rate was 10 °C/min.

### 2.2. Preparation of High Entropy Alloy Nanopowder Composites and Characterization of Soft Magnetic Properties

Lake Shore 7300 vibrating sample magnetometer was used to test the hysteresis lines of high entropy alloy nanopowders. An Agilent E8363 vector network analyzer was used to measure the electromagnetic parameters of the specimen in the frequency range of 0.3–8.5 GHz using the coaxial method. The sample was a coaxial toroidal ring with an inner diameter of Φ3 mm and an outer diameter of Φ7 mm, consisting of 85 wt% alloy powders and 15 wt% paraffin wax. The inductance L and quality factor Q of the magnetic powder cores at different frequencies were tested by using Tonghui TH2816B digital bridge, and the DC bias performance of the magnetic powder cores was tested by using Tonghui TH2816B digital bridge and TH1778 DC bias current source, and the outer diameter of the annular magnetic powder core samples was Φ20 mm, the inner diameter of the annulus Φ10 mm, and the height of the annulus was 5.5 mm.

## 3. Preparation Process, Powder Morphology, Structure and Properties of High Entropy Alloy Nanopowders

The optimized process for the preparation of high entropy alloy nanopowders was [N_2_H_4_]/([Ni^2+^] + [Co^2+^] + [Cr^3+^] + [Fe^2+^] + [Y^3+^]) > 2/1 (molar ratio), the reaction starting pH = 13, the reaction temperature was 80 °C and the reaction time was 40 min. The reaction was mainly affected by factors such as the temperature, time, concentration of the reducing agent, and the dispersant. Figure 1a showed the reaction product specimens at 5 min intervals for 40 min of reaction, and Figure 1b shows the XRD results of the products during the reaction. Figure 1b showed that at the beginning of the reaction, the intermediate products such as FeOOH, CoOOH, NiOOH, CrOOH, Y(OH)_3_, YH_3_, etc. were formed first, and the FeOOH and CoOOH products basically disappeared at 15 min. At the same time, as the reaction proceed, [Ni^2+^], [Co^2+^], [Cr^3+^] induced [Fe^2+^] and [Y^3+^] reduction to generate FeCr, FeNi_3_, NiCrFe, Ni_9_Y, Cr_2_Ni_3_, FeCo and other alloy phases, the PH value also gradually decreased from 13 to 10. Meanwhile, according to the Scherrer formula, the crystal particle size at 5 min was 0.472 nm, at 10 min it was 0.41 nm, at 15 min it was 0.508 nm, and at 25 min it was 0.855 nm.

The prepared high entropy alloy powders were black (Figure 2a), and the test of the high entropy alloy powder close to a permanent magnet showed that the powder already has soft magnetic properties. The particle size of the powders were about 100 nm, spherical (Figure 2b). Further EDS surface scanning (Figure 2c) showed that the powders were composed of Ni, Fe, Co, Cr, and Y elements, with some powders mainly composed of Fe and Ni, and the other part consisting of Fe, Ni, Co, Cr, and Y. This was because the Ni content was higher than that of other elements, so Fe and Ni first reacted to form FeNi_3_ phase. As the Ni content decreased to a concentration similar to that of other elements, small particle size powders were generated as FeNiCoCrY alloy powders. In addition, the maximum equilibrium phase formula given by the Gibbs phase rule is *p* = *n* + 1, where *p* represents the number of coexisting phases in equilibrium and *n* represents the number of independent components in the alloy. The formula for non-equilibrium states is *p* > *n* + 1, while for high entropy alloys, the number of phases obtained is much smaller than the number of components in the alloy (i.e., *p* < *n* + 1). In this article, the *p*-value was 2 and the *n*-value was 5, indicating the successful preparation of high entropy alloys.

XRD analysis showed (Figure 2d) that the diffraction peaks at 2θ = 44.2°, 51.5° and 75.8° of the high entropy alloy powders after the completion of the reaction, corresponding to the (111), (200), and (220) crystal planes, respectively, represent the FeNi_3_ phase. Diffraction peaks at 2θ = 44.6°, 65.0° and 82.3° correspond to (110), (200) and (211) crystal faces for the CrFe_4_ phase, respectively. Diffraction peaks at 2θ = 44.7°, 65.1° and 82.4°, corresponding to (111), (200) and (211) crystal planes, respectively, represent the Co_3_Fe_7_ phase. Diffraction peaks at 2θ = 43.7°, 51.0° and 75.0°, corresponding to (111), (200) and (220) crystal faces, represent the Cr_2_Ni_3_ phase. Diffraction peaks at 2θ = 30.9°, 37.2°, 43.7° and 49.1°, corresponding to (101), (110), (111) and (201) crystal planes, respectively, represent the Co_5_Y phase. The diffraction peaks at 2θ = 44.6°, 64.9°and 82.3°, corresponding to (111), (200) and (211) crystal faces, represent the NiCrFe phase. Diffraction peaks at 2θ = 35.4°, 43.7° and 67.7°, corresponding to (220), (222) and (511) crystal faces, represent the Co_2_Y phase. The diffraction peaks at 2θ = 35.4°, 43.8° and 67.9°, corresponding to the (220), (222) and (511) crystal planes, respectively, represent the Ni_2_Y phase. Thus, the main phase structures of alloy powders were the FeNi_3_ phase, CrFe_4_ phase, Co_3_Fe_7_ phase, Cr_2_Ni_3_ phase, Co_5_Y phase, NiCrFe phase, Co_2_Y phase and Ni_2_Y phase.

The DSC curves show (Figure 2e) that the high entropy alloy nanopowders have high thermal stability below 100 °C. The thermal decomposition process of NiCoCrFeY powders mainly was follows: There was an obvious exothermic peak at 100 °C, indicating heat release, and the powders subsequently started to decompose and oxidize. The specific gravity of the powders noticeably decreased with increasing temperature, reaching the maximum loss at 550 °C. After this point, the powders’ quality became stable, and the oxidation process stopped. The XPS full spectrum of the powders showed that Ni, Fe, Co, Cr, Y and other elements also existed in the prepared samples (Figure 2f).

## 4. Soft Magnetic Properties of High Entropy Alloy Nanopowders

The magnetization curve of the powders showed that the saturation magnetization field strength of the powders were up to 5000 Qe, and the saturation magnetization strength Ms was about 17.3 emu/g. It has good soft magnetic properties and was suitable to be applied in high intensity magnetic fields and high frequency magnetic fields. Figure 3b showed the H-B curve of the high entropy alloy nanopowders, the coercive force Hc of high entropy alloy nanopowders were 250 A/m, the remanent magnetic Br value were 0.5 mT, and the saturated magnetic induction BS value were 9 mT.

### 4.1. Electromagnetic Properties of High Entropy Alloy Nanopowders

As shown in Figure 4, the real part of the complex permittivity, ε′, of nanopowders was about 6.7–8.14, the real part of the complex permeability, μ′, of nanopowders was about 1.03–1.51, the tgδε value was 0.11–0.13, and the tgδμ value was 0.08–0.15. Compared with the alloy nanopowders, the high entropy alloy nanopowders have a very low value of tgδε and tgδμ value, which was close to the level of the pure resin, indicating that this powder has the excellent property of low high frequency loss [37].

The shielding and loss benefits of powder on electromagnetic waves can be characterized by transmission line theory. Generally, when the reflection coefficient loss R value is less than −10 dB, it means that 90% of the incident electromagnetic wave can be lost, which is considered to be effectively absorbed, and the frequency band less than −10 dB is regarded as the effective absorption band (EAB). Through the electromagnetic parameter fitting, at 0.3–18 GHz, the strongest absorption peak (−14 dB) occurred at 16.7 GHz when the thickness of the powders coating was 5 mm. The bandwidth of the effective absorption peak reached 2.5 GHz, which showed excellent absorption performance in the Ku band. The strongest absorption peak (−15.6 dB) occurred at 8.4 GHz when the thickness of the powders coating was 10 mm; The effective absorption peak bandwidth reached 1.2 GHz, which showed good wave-absorbing performance in the X-band.

### 4.2. Inductance Properties of High Entropy Alloy Nanopowders

Magnetic powder core is a typical soft magnetic composite material. The metal magnetic powder core is a soft magnetic material pressed by mixing magnetic material powder (usually iron-based powder) and an insulating medium, which has high permeability, high resistivity, low loss and magnetic isotropy. Due to its ability to be used at higher frequencies and with higher power, it has advantages in many applications that are difficult to compare with other magnetic materials and has been widely used in inductive filters, choke coils, and switching power cores in telecommunications, radar, television, and power supply technologies.

The treatment of the insulating cladding is the main factor affecting the effective permeability of the magnetic powder core and its loss. The coating agent content has a strong influence on the equivalent permeability and magnetic losses of soft magnetic composites. When the coating agent content in the magnetic powder core is 0.4% to 4.0% (mass ratio with iron-based powder), the magnetic loss of the magnetic powder core gradually increases with the increase of frequency, while the equivalent permeability decreases significantly. In contrast to iron-based powder magnetic cores, high entropy alloy nanopowders have small grain sizes and a much larger interfacial area between the grains than micron-sized powders, so magnetic cores do not require a cladding treatment. In addition, the powders have good moulding, and the powders have large specific surface area, so that the magnetic powder core composites could be prepared under low pressure and without coating insulators. High entropy alloy nanopowders have the same plasticity as pure iron powder, and the pressing pressure is ≤500 Mpa, which is significantly lower than that of iron-based powder magnetic powder core (≥1000 Mpa).

The heat treatment process of the magnetic powder core is generally carried out under the protection of nitrogen at 300–800 °C, mainly to reduce the residual stresses and defects left when pressing the encapsulated iron-based powder. According to the DSC curve, the high entropy alloy nanopowders were heat-treated at 550 °C × 1 h and 800 °C × 1 h under nitrogen protection. The magnetic powder core shrank in volume and increased in density after thermal treatment. The results of the oxidation process are shown in Figure 5d.

The high entropy alloy nanopowder magnetic powder cores showed excellent frequency-constant magnetic properties and magnetic field-constant magnetic properties. In the frequency range from 1 KHz to 100 KHz, the starting permeability μi and maximum permeability µm of the magnetic powder core heat-treated at 800 °C were 17 and 359 (Table 1), respectively, and μ_e_ was about 4.6, and the variation of μ_e_ with frequency was less than 1% (Figure 5c); inside the magnetic powder core, the magnetic lines of force were uniformly distributed, for example, when the applied magnetic field H was (0.1 A current, 20-turned line) 42 A/m, the Φ20-10-10 magnetic powder core’s inner surface B-value and outer surface B-value were 0.17 mT and 0.24 mT, respectively, and the difference between the inner and outer surface B-values was much lower than that of alloy materials, which was attributed to the high overall electrical resistivity of the nanopowder soft magnetic materials, and the material exhibits bad conductor properties.

Magnetic powder cores maintained a constant μ_e_ value under the action of the external magnetic field from 0 to 12,000 A/m. According to the constant magnetic properties of the magnetic powder core soft magnetic composites under the action of the high external magnetic field, they can be used as air gap filling materials for soft magnetic alloy materials. Compared with the commonly used air gap, the high entropy alloy nanocomposites can effectively increase the effective permeability of the alloy cores. Taking a typical FeCuNbSiB nanocrystalline magnetic core as a case study, the core specification is Φ20 mm outer diameter, Φ10 mm inner diameter, 10 mm height, 20 coil turns, 1A coil current, and the FeCuNbSiB nanocrystalline alloy has a μ_e_ of 60,000 at a frequency of 1 KHz. The air gap ratio (the ratio of the width of the air gap d to the periphery of the centre of the core, in %) was at 1%, 3%, and 5%. The high entropy alloy nanocomposites air gap material can effectively increase the effective permeability of the alloy compared with the commonly used air gap materials. Entropy alloy nanocomposite air gapped cores reached μ_e_ values of 164, 60, and 38, respectively, which were 4, 3.2, and 3 times higher than those of air gapped cores (Figure 6c).

## 5. Conclusions

High entropy alloy nanopowders were successfully prepared by liquid-phase reduction methods. The prepared high entropy alloy nanopowders consisted of FeNi alloy spherical powders and NiFeCoCrY alloy spherical powders with a particle size of about 100 nm. EDS surface scanning showed that the powders were composed of Ni, Fe, Co, Cr, and Y elements, with some powders mainly composed of Fe and Ni, and the other part consisting of Fe, Ni, Co, Cr, and Y.

High entropy alloy nanopowders showed that the saturation magnetization field strength of the powders were up to 5000 Qe, and the saturation magnetization strength Ms was about 17.3 emu/g. It has good soft magnetic properties and was suitable to be applied in high intensity magnetic fields and high frequency magnetic fields. At 0.3–18 GHz, the strongest absorption peak (−14 dB) occurred at 16.7 GHz when the thickness of the powders coating was 5 mm. The bandwidth of the effective absorption peak reached 2.5 GHz, which showed excellent absorption performance in the Ku band. The strongest absorption peak (−15.6 dB) occurred at 8.4 GHz when the thickness of the powders coating was 10 mm; The effective absorption peak bandwidth reached 1.2 GHz, which showed good wave-absorbing performance in the X-band.

NiCoCrFeY high entropy alloy nanopowders have good plasticity and large specific surface area of the powder, which can be used to prepare magnetic powder core composites under low pressure moulding and without coating insulators, and the magnetic powder cores showed excellent frequency-constant magnetic properties and magnetic field-constant magnetic properties. In the frequency range of 1~100 KHz, the μ_m_ of the 800 °C heat-treated magnetic powder core reached 359, the μ_e_ was about 4.6, and the change rate of μ_e_ with frequency was less than 1%; the magnetic powder core maintained a constant μ_e_ value under the action of the external magnetic field from 0 to 12,000 A/m.

## Figures and Tables

**Figure 1 materials-17-00534-f001:**
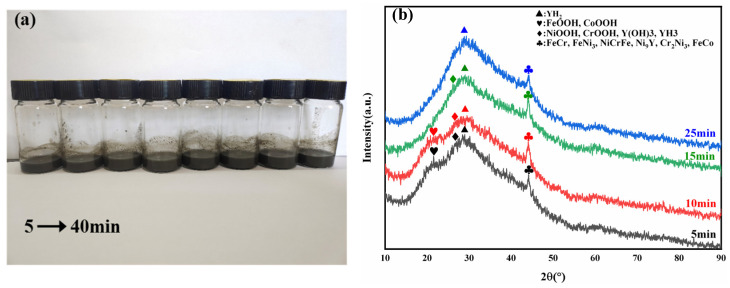
Reaction process of high entropy alloy nanopowders: (**a**) 5–40 min reaction process solution; (**b**) powders XRD diagram of 25 min reaction.

**Figure 2 materials-17-00534-f002:**
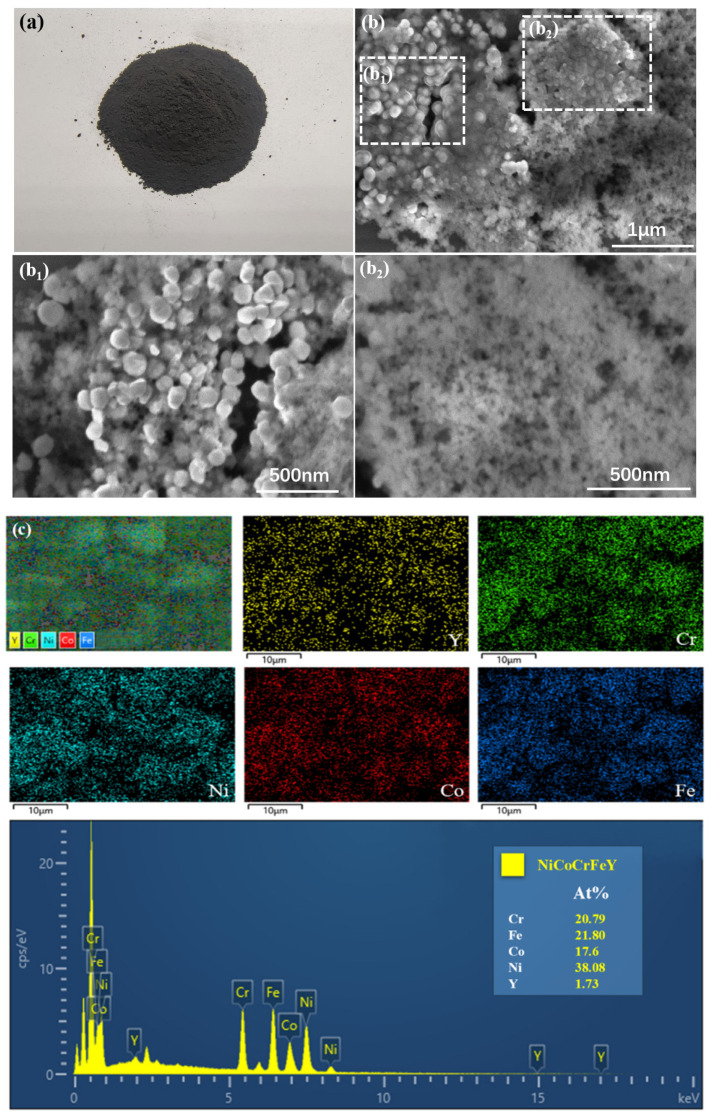
(**a**) High entropy alloy nanopowders; (**b**) SEM diagram of high entropy alloy nanopowders (Enlarged illustrations of **b1** and **b2** as two alloy phases of **b**); (**c**) Element distribution diagram and EDS diagram of high entropy alloy nanopowders; (**d**) XRD diagram of high entropy alloy nanopowders produced by reaction; (**e**) DSC diagram of high entropy alloy nanopowders; (**f**) XPS diagram of high entropy alloy nanopowders.

**Figure 3 materials-17-00534-f003:**
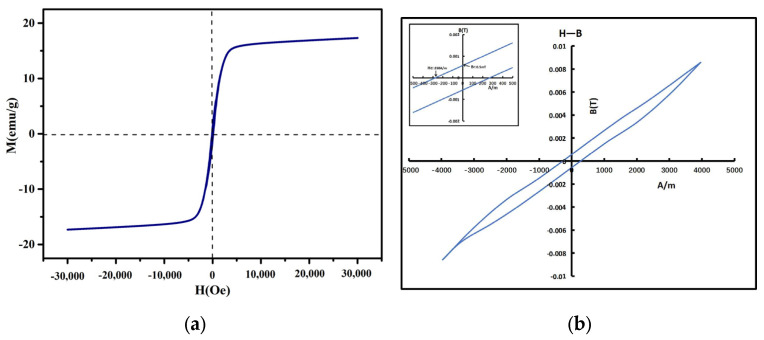
(**a**) Magnetic saturation curve Ms diagram of high entropy alloy nanopowders; (**b**) H-B curve diagram of high entropy alloy nanopowders.

**Figure 4 materials-17-00534-f004:**
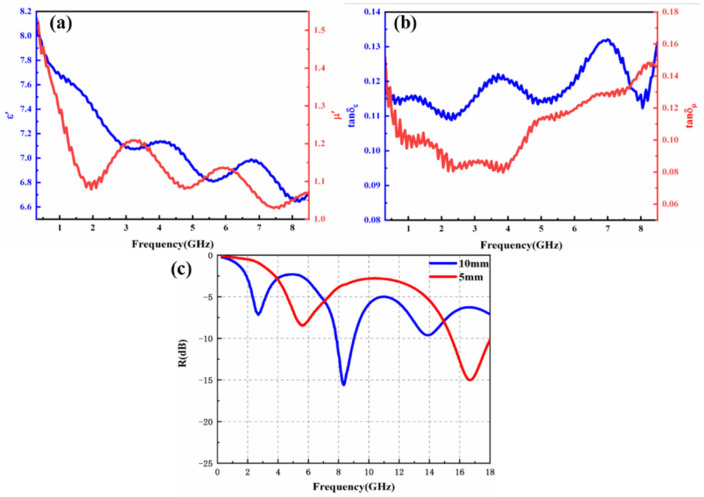
Electromagnetic parameters of high entropy alloy nanopowders: (**a**) ε′ and μ′ of powders; (**b**) tgδε and tantgδμ of powders; (**c**) wave-absorbing properties of composite powder coatings.

**Figure 5 materials-17-00534-f005:**
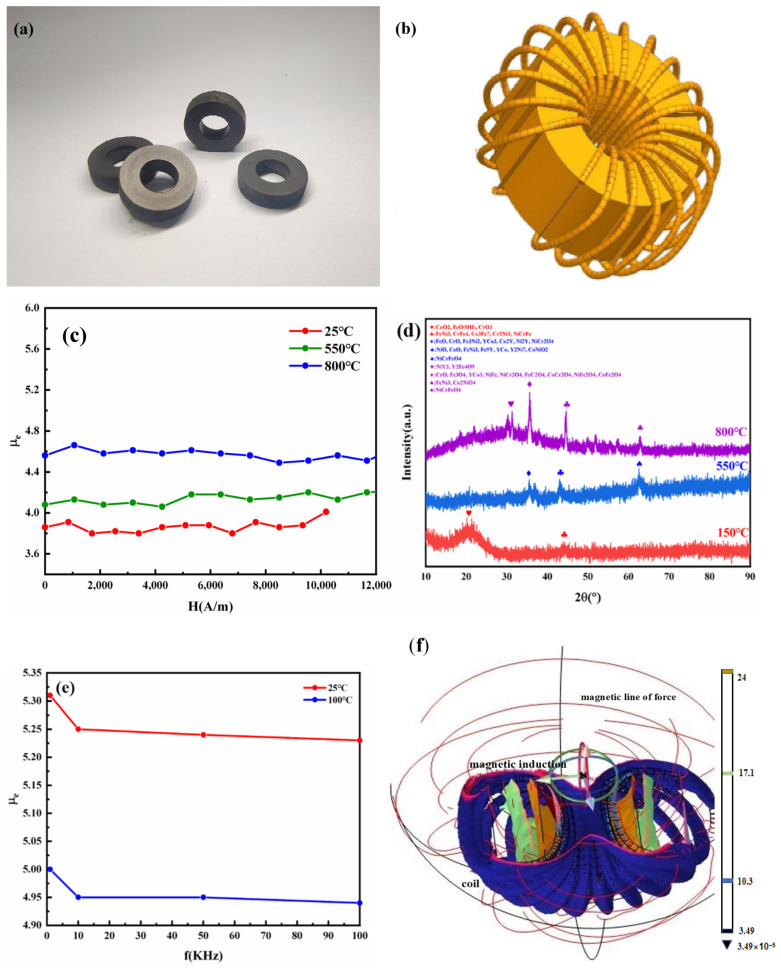
(**a**) Magnetic powder core; (**b**) Inductance; (**c**) μ_e_ values at different heat treatment temperatures; (**d**) XRD values at different temperatures; (**e**) μ_e_ values at different frequencies; (**f**) B-value distribution on the inner and outer surfaces of the core.

**Figure 6 materials-17-00534-f006:**
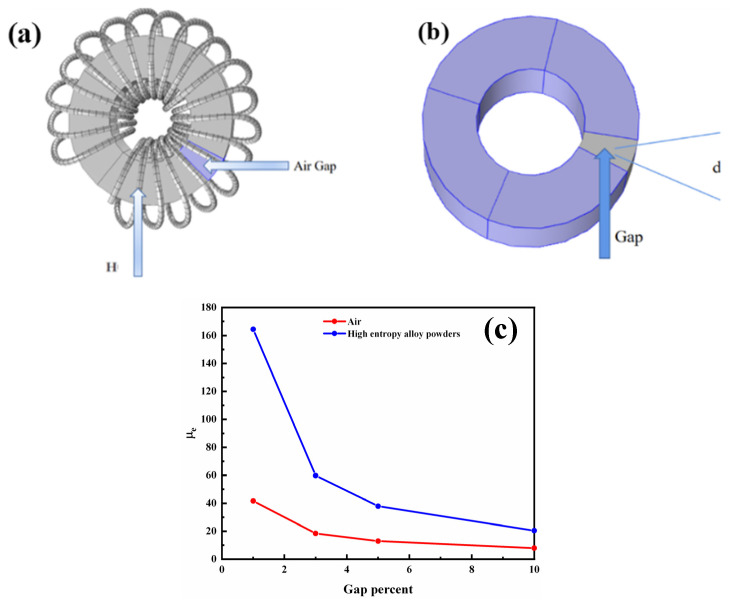
(**a**) Inductance with air gap; (**b**) core with air gap; (**c**) comparison of μ_e_ values between the composite air gap and air gap cores.

**Table 1 materials-17-00534-t001:** Soft magnetic properties of powders.

Temperature (°C)	μ_i_ (k)	μ_m_ (k)	Pu (GOe)	Bs (mT)	Br (mT)	Hc (A/m)	Br/Bs
25	0.020	0.237	711.2	8.472	1.012	54.31	0.119
550	0.019	0.850	503.9	6.274	0.224	325.7	0.036
800	0.017	0.359	1679	12.41	1.729	223.9	0.139

## Data Availability

Exclude this statement if the study did not report any data.

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
