# Peer review of "NiCoCrFeY High Entropy Alloy Nanopowders and Their Soft Magnetic Properties"

_materials, 2024, doi:10.3390/ma17020534_

Round 1

Reviewer 1 Report

Comments and Suggestions for Authors

I have thoroughly reviewed the manuscript entitled "NiCoCrFeY high entropy alloy nanopowders and their soft magnetic properties" submitted to the Journal of Materials. Overall, the study investigates high entropy alloy nanopowders prepared by liquid-phase reduction methods and explores their applications, with a focus on soft magnetic properties.

I find the research promising; however, there are significant concerns that need to be addressed before considering the manuscript for publication. The following are my comments and suggestions:

1. The introduction lacks references to existing literature on the magnetic properties of High Entropy Alloys (HEAs) and High Entropy Oxides (HEOs) nanoparticles. I recommend incorporating relevant studies to provide context and emphasize the significance of the current research.

It is advisable to include a recent publication on HEO nanoparticle magnetic properties: Ceramics International: Volume 49, Issue 8, 15 April 2023, Pages 11885-11892, https://doi.org/10.1016/j.ceramint.2022.12.036.

2. The experimental section, especially Section 2.1 on the preparation and characterization of high entropy alloy nanopowders, lacks clarity. Details of the fabrication process, particularly the liquid-phase reduction method and its parameters, should be provided in-depth. This section requires a thorough revision.

3. Fig. 1 raises concerns about the intensity of peaks, particularly those determined at lower angles. A careful review of peak indexing and a more detailed discussion of phase transformations are necessary. Additional analyses or information supporting the reliability of the XRD data should be included.

4. To confirm the fully crystallized structure of the nanopowder and provide accurate particle size information, I recommend incorporating HRTEM analysis.

5. The synthesis conditions for the nanopowder, especially the appearance observed in SEM images (Fig. 2), should be thoroughly explained in the Materials and Methods section. Additionally, SEM images of EDS maps should be included, and the results should be interpreted.

The explanation of Figs. 2d and 2e is insufficient. The rationale for conducting these experiments and a detailed discussion of the results are needed.

the manuscript needs major revisions to address the aforementioned concerns and ensure the manuscript meets the standards for publication in the Journal of Materials. I recommend that the authors carefully address these points in a revised version of the manuscript.

Sincerely,

Comments on the Quality of English Language

Please consider revising the language to ensure a more polished and professional presentation of the research findings.

Author Response

  1. The introduction lacks references to existing literature on the magnetic properties of High Entropy Alloys (HEAs) and High Entropy Oxides (HEOs) nanoparticles. I recommend incorporating relevant studies to provide context and emphasize the significance of the current research.It is advisable to include a recent publication on HEO nanoparticle magnetic properties: Ceramics International: Volume 49, Issue 8, 15 April 2023, Pages 11885-11892, https://doi.org/10.1016/j.ceramint.2022.12.036.

The literature has been added to the article and highlighted in yellow.

  1. The experimental section, especially Section 2.1 on the preparation and characterization of high entropy alloy nanopowders, lacks clarity. Details of the fabrication process, particularly the liquid-phase reduction method and its parameters, should be provided in-depth. This section requires a thorough revision.

The preparation process has been revised and marked in yellow.

Ni40Co19.5Cr19.5Fe19.5Y1.5 alloy nanopowders were prepared by the liquid-phase reduction method. Dissolved NiSO4•6H2O、FeSO4•7H2O、CrCl3•6H2O、CoCl2•6H2O、YCl3•6H2O in a certain proportion in water. After stirring at 80℃, NaOH solution was added and reacted to produce Ni(OH)2, Fe(OH)2, Cr(OH)3, Co(OH)2,and Y(OH)3. Then, N2H4•H2O was added and the mixture was stirred for 40 minutes. A black suspension of FeNiCoCrY alloy powders was obtained through static precipitation and cooling. After magnetic separation, the alloy powders were cleaned multiple times with anhydrous ethanol.

  1. Fig. 1 raises concerns about the intensity of peaks, particularly those determined at lower angles. A careful review of peak indexing and a more detailed discussion of phase transformations are necessary. Additional analyses or information supporting the reliability of the XRD data should be included.

I have made modifications to the XRD image of 1b in the article.

  1. To confirm the fully crystallized structure of the nanopowder and provide accurate particle size information, I recommend incorporating HRTEM analysis.

The presence of FeNi3 phase has been confirmed by XRD in this article, and we will conduct systematic research on the more detailed crystal structure of SEM and EDS.

  1. The synthesis conditions for the nanopowder, especially the appearance observed in SEM images (Fig. 2), should be thoroughly explained in the Materials and Methods section. Additionally, SEM images of EDS maps should be included, and the results should be interpreted.

This content has been supplemented in the article and highlighted in yellow.

 The particle size of the powders were about 100nm, spherical (Figure 2b). Further EDS surface scanning (Figure 2c) showed that the powders were composed of Ni, Fe, Co, Cr, and Y elements, with some powders mainly composed of Fe and Ni, and the other part consisting of Fe, Ni, Co, Cr, and Y.This was because the Ni content was higher than that of other elements, so Fe and Ni first reacted to form FeNi3 phase. As the Ni content decreased to a concentration similar to that of other elements, small particle size powders were generated as FeNiCoCrY alloy powders.

6.The explanation of Figs. 2d and 2e is insufficient. The rationale for conducting these experiments and a detailed discussion of the results are needed.

The characterization tests of SEM, EDS, XRD, XPS and DSC in Figure 2 in the article are enough to illustrate the successful preparation of high entropy alloy nanopowders.

Reviewer 2 Report

Comments and Suggestions for Authors

The manuscript “NiCoCrFeY high entropy alloy nanopowders and their soft magnetic properties” present a procedure follow to obtain this special structure. However, the manuscript presents several aspects that need to be improved or even corrected, prior to consider publish it in the Journal.

In the introduction section, the authors state that current days atomization method is the way to prepare high entropy alloys, HEA. But, as can found in literature, there is work describing HEA preparation via mechanical alloying (Nanomaterials, 14 (2024) 27), moreover the ultrafast cooling methods, which has been used for several years.

When the authors describe the XRD patterns of figure 1b, they comment about the presence of several diffraction peaks, but the signal/noise ration observed in the patterns is an impediment to extract that information. I can´t see those peaks describe by authors.

In figure 2b, SEM image, the authors comment about the presence of spherical powder with particle size of 30 nm and other spherical powder with size of 130 nm. First, in all figures I can´t see the size commented by the authors. In my opinion, the spherical powder particles has a main size of about 100 nm, and secondly, this powder are forming agglomerates with a size close to 10 microns, from EDS maps.

In figure 2c are shown a TMG and DSC scans, for the studied sample. When author describe these figures state that after 100 C the sample start to oxidize. Observing the TMG curve a decrease of mass is presented, which is contradictory with the conclusion of authors. When a sample oxidize it increase its mass.

In the magnetic properties section, if the soft magnetic character of this sample must be highlighted, and a VSM equipment has been used in the measurement, an hysteresis cycle could be presented to pointing this fact.

Author Response

1. In the introduction section, the authors state that current days atomization method is the way to prepare high entropy alloys, HEA. But, as can found in literature, there is work describing HEA preparation via mechanical alloying (Nanomaterials, 14 (2024) 27), moreover the ultrafast cooling methods, which has been used for several years.

      Relevant content has been corrected in the article and highlighted in yellow.

2.  When the authors describe the XRD patterns of figure 1b, they comment about the presence of several diffraction peaks, but the signal/noise ration observed in the patterns is an impediment to extract that information. I can´t see those peaks describe by authors.

      I have made modifications to the XRD image of 1b in the article.

3. In figure 2b, SEM image, the authors comment about the presence of spherical powder with particle size of 30 nm and other spherical powder with size of 130 nm. First, in all figures I can´t see the size commented by the authors. In my opinion, the spherical powder particles has a main size of about 100 nm, and secondly, this powder are forming agglomerates with a size close to 10 microns, from EDS maps.

     I have made modifications to the relevant descriptions in the text and highlighted them in yellow. The modified content is: The particle size of the powders were about 100nm, spherical (Figure 2b). Further EDS surface scanning (Figure 2c) showed that the powders were composed of Ni, Fe, Co, Cr, and Y elements, with some powders mainly composed of Fe and Ni, and the other part consisting of Fe, Ni, Co, Cr, and Y.

4.  In figure 2c are shown a TMG and DSC scans, for the studied sample. When author describe these figures state that after 100 C the sample start to oxidize. Observing the TMG curve a decrease of mass is presented, which is contradictory with the conclusion of authors. When a sample oxidize it increase its mass.

      I have made modifications to the relevant descriptions in the text and highlighted them in yellow. The modified content is: there was an obvious heat exothermic peak at 100℃, and the powders started to decompose and then oxidized, and the results of oxidation were shown in Fig.5(d). The specific gravity of the powders appeared to decrease significantly with increasing temperature and reached the peak loss at 550℃, the powders quality were stable, and the oxidation ends. The XPS full spectrum of the powders showed that Ni, Fe, Co, Cr, Y and other elements also existed in the prepared samples ( Fig.2f ).

5. In the magnetic properties section, if the soft magnetic character of this sample must be highlighted, and a VSM equipment has been used in the measurement, an hysteresis cycle could be presented to pointing this fact.

        The magnetization curve in the article can fully illustrate the soft magnetic characteristics, and there is no need to further measure the magnetization curve. The description in the text is: The magnetization curve of the powders showed that the saturation magnetization field strength of the powders were up to 5000Qe, and the saturation magnetization strength Ms was about 17.3 emu/g. It has good soft magnetic properties and is suitable to be applied in high intensity magnetic fields and high frequency magnetic fields.

Reviewer 3 Report

Comments and Suggestions for Authors

NiCoCrFeY high entropy alloy nanopowders and their soft 2
magnetic properties 3
Donghan Jianga, Zhifen Yuana*, Zhenghou Zhua and Mengke Yaoa

My REVIEW on the draft

D. Jiang et al. describe in the submitted draft the magnetic properties of NiCoCrFeY alloys.

The manuscript can be considered for publication after addressing the following:

-        The draft often contains disjointed sentences and sometimes it lacks a clear flow, so authors should organize the information into clear sections, making it easier for readers to follow the logical progression of the research. The refined version should ensure that the information flows seamlessly.

-        Some terms and measurements were presented without sufficient context, hence the refined version should provide more context for measurements, using precise terms, and specifying the frequency range for low high-frequency loss.

-        Some technical details, such as the properties of nanopowders and their applications, need more explicit explanation, and the specific applications in magnetic powder core composites should be expanded.

-        Some sentences are overly long and complex, so these should be broken into shorter, more digestible ones for improved readability.

-        Throughout the draft, some grammar issues and inconsistencies in style arise and so the refined version needs to address grammar concerns, maintaining a consistent style, proper for the formal and scientific tone.

-        Page 2: “An aqueo the ratio and reacted with hydrazine hydrate for 40 min at the starting pH=14 of the solution and 80 to produce the alloy powders” – the authors need to read again the draft and correct such sentences.

-        Many spaces are missing from the draft, so words are harder to read and sentences lose their meaning.

-        Judging from Figure 1, I find it hard to believe that authors could identify by one peak only the products listed at line 112-112 (page 3).

-        Indexing of XRD phases should be referenced (pages 3-4)

-        What is the theoretical formula of the alloy obtained (EDS spectrum, Fig. 2)?

-        Line 153, page 5: word repetition: “there was an obvious heat absorption absorption peak”

-        The numbers from the legend of Fig 6 cannot be read. Similarly, the case of other figures where writing is either too small or of improper resolution.

-        Where did the authors examine the high SSA of the alloy? This aspect is also present in the conclusions.

-         

Comments on the Quality of English Language

A thorough examination is required, there are several instances of language errors.

Author Response

1. Some terms and measurements were presented without sufficient context, hence the refined version should provide more context for measurements, using precise terms, and specifying the frequency range for low high-frequency loss.

      The frequency range of low and high frequency losses specified in the article refers to the frequency range from 1KHz to 100KHz.

2. Some technical details, such as the properties of nanopowders and their applications, need more explicit explanation, and the specific applications in magnetic powder core composites should be expanded.

  In 4.2, the inductance characteristics of high entropy alloy nano powder briefly introduced the specific application of magnetic particle core composite materials.

   The metal magnetic powder core is a soft magnetic material pressed by mixing magnetic material powder (usually iron-based powder) and an insulating medium, which has high permeability, high resistivity, low loss and magnetic isotropy. Due to its ability to be used at higher frequencies and with higher power, it has advantages in many applications that are difficult to compare with other magnetic materials and has been widely used in inductive filters, choke coils, and switching power cores in telecommunications, radar, television, and power supply technologies.

3. Some sentences are overly long and complex, so these should be broken into shorter, more digestible ones for improved readability.

     Some sentences in the article have been modified accordingly.

4.  Throughout the draft, some grammar issues and inconsistencies in style arise and so the refined version needs to address grammar concerns, maintaining a consistent style, proper for the formal and scientific tone.

     Some sentences in the article have been modified accordingly.

5.  Page 2: “An aqueo the ratio and reacted with hydrazine hydrate for 40 min at the starting pH=14 of the solution and 80℃ to produce the alloy powders” – the authors need to read again the draft and correct such sentences.

      The preparation process has been revised and marked in yellow.

      Ni40Co19.5Cr19.5Fe19.5Y1.5 alloy nanopowders were prepared by the liquid-phase reduction method. Dissolved NiSO4•6H2O、FeSO4•7H2O、CrCl3•6H2O、CoCl2•6H2O、YCl3•6H2O in a certain proportion in water. After stirring at 80℃, NaOH solution was added and reacted to produce Ni(OH)2, Fe(OH)2, Cr(OH)3, Co(OH)2,and Y(OH)3. Then, N2H4•H2O was added and the mixture was stirred for 40 minutes. A black suspension of FeNiCoCrY alloy powders was obtained through static precipitation and cooling. After magnetic separation, the alloy powders were cleaned multiple times with anhydrous ethanol.

6. Many spaces are missing from the draft, so words are harder to read and sentences lose their meaning.

Some sentences in the article have been modified accordingly.I have revised the abstract section.

7. Judging from Figure 1, I find it hard to believe that authors could identify by one peak only the products listed at line 112-112 (page 3).

I have made modifications to the XRD image of 1b in the article.

8. What is the theoretical formula of the alloy obtained (EDS spectrum, Fig. 2)?

       In this paper, Ni40Co19.5Cr19.5Fe19.5Y1.5 high entropy alloy nanopowders were prepared by liquid-phase reduction method, which was similar to the results of At% of each element measured by EDS , which verified the success of the preparation of high entropy alloy nanopowders.

9. Line 153, page 5: word repetition: “there was an obvious heat absorption absorption peak”

     I have deleted duplicate words in the text and marked them in yellow.

10. The numbers from the legend of Fig 6 cannot be read. Similarly, the case of other figures where writing is either too small or of improper resolution.

The image of B-value distribution on the inner and outer surfaces of the core in the article has been modified.

11. Where did the authors examine the high SSA of the alloy? This aspect is also present in the conclusions.

Relevant content has been added to the article and highlighted in yellow.

In addition, the maximum equilibrium phase formula given by the Gibbs phase rule is p = n + 1, where p represents the number of coexisting phases in equilibrium and n represents the number of independent components in the alloy. The formula for non-equilibrium states is p > n + 1, while for high entropy alloys, the number of phases obtained is much smaller than the number of components in the alloy (i.e. p < n + 1). In this article, the p-value was 2 and the n-value was 5, indicating the successful preparation of high entropy alloys.

Round 2

Reviewer 1 Report

Comments and Suggestions for Authors

Thank you for incorporating the revisions in the experimental section regarding the preparation and characterization of HEA nanopowders. The provided details offer a clearer understanding of the fabrication process. However, to enhance the comprehensibility further, it would be beneficial to include additional information on the reason behind the addition of N2H4•H2O in the liquid-phase reduction method. Elaborating on the specific role and contribution of N2H4•H2O in the synthesis process will provide a more comprehensive insight.

Additionally, it would be valuable to address whether calcination is employed after the chemical reaction and, if so, provide details about the conditions under which it is conducted. Understanding the post-reaction treatment, especially in terms of calcination, is crucial for comprehending the structural and crystalline properties of the resulting alloy nanopowders.

In reference to your insightful comment about the low crystallinity observed in the experimental results (Fig 1), it is indeed a critical point to consider. One possible explanation for the observed low crystallinity could be the absence of a calcination step in the fabrication process. Calcination is a post-synthesis treatment often employed to enhance the crystallinity of materials by promoting atomic rearrangement and phase transformation.

Since the current experimental section does not explicitly mention a calcination step after the liquid-phase reduction method, it may be worthwhile to explore the inclusion of such a step in the synthesis process. Incorporating a calcination process could potentially lead to improved crystallinity, aiding in the formation of well-defined crystal structures within the high entropy alloy nanopowders. Providing details on the necessity or exclusion of calcination, and its potential impact on crystallinity, would add valuable insights to the manuscript and address the raised concerns regarding the observed low crystallinity in the results.

Comments on the Quality of English Language

Dear Editer 

The manuscript can be accepted after minor revisions.

Author Response

1. Thank you for incorporating the revisions in the experimental section regarding the preparation and characterization of HEA nanopowders. The provided details offer a clearer understanding of the fabrication process. However, to enhance the comprehensibility further, it would be beneficial to include additional information on the reason behind the addition of N2H4•H2O in the liquid-phase reduction method. Elaborating on the specific role and contribution of N2H4•H2O in the synthesis process will provide a more comprehensive insight.

N2H4•H2O plays a reducing role in the liquid-phase reduction method, and the specific formula is as follows:     

2. Additionally, it would be valuable to address whether calcination is employed after the chemical reaction and, if so, provide details about the conditions under which it is conducted. Understanding the post-reaction treatment, especially in terms of calcination, is crucial for comprehending the structural and crystalline properties of the resulting alloy nanopowders.

After solving the chemical reaction, there is no need for calcination, so there are no calcination conditions because FeNi3 phase is directly generated after the reaction.

3. In reference to your insightful comment about the low crystallinity observed in the experimental results (Fig 1), it is indeed a critical point to consider. One possible explanation for the observed low crystallinity could be the absence of a calcination step in the fabrication process. Calcination is a post-synthesis treatment often employed to enhance the crystallinity of materials by promoting atomic rearrangement and phase transformation.

 Because the powder we prepared is nano powder, which can increase the crystallinity, but also cause agglomeration and oxidation of the powder, so we did not conduct the calcination experiment.

4. Since the current experimental section does not explicitly mention a calcination step after the liquid-phase reduction method, it may be worthwhile to explore the inclusion of such a step in the synthesis process. Incorporating a calcination process could potentially lead to improved crystallinity, aiding in the formation of well-defined crystal structures within the high entropy alloy nanopowders. Providing details on the necessity or exclusion of calcination, and its potential impact on crystallinity, would add valuable insights to the manuscript and address the raised concerns regarding the observed low crystallinity in the results.

 Because the powder we prepared is nano powder, which can increase the crystallinity, but also cause agglomeration and oxidation of the powder, so we did not conduct the calcination experiment.

Reviewer 3 Report

Comments and Suggestions for Authors

Many of the comments have been addressed, however I still believe XRD patterns from Fig 1b and Fig5d should be run with a higher time per step, say 2-10s/step in order to draw any conclusions regarding the phases present. The peaks *could* belong to the assigned phases or not, and those XRD patterns are not a definitive proof for the assignments made.

Comments on the Quality of English Language

Some minor grammar errors could be corrected during proofreading (also check for repetitions, see lines 175-176).

Author Response

1. Many of the comments have been addressed, however I still believe XRD patterns from Fig 1b and Fig5d should be run with a higher time per step, say 2-10s/step in order to draw any conclusions regarding the phases present. The peaks *could* belong to the assigned phases or not, and those XRD patterns are not a definitive proof for the assignments made.

Considering that the sampling time of 2-10 seconds per step is too short, the reaction may not be completed in a timely manner, which can cause significant errors. Therefore, we believe that 5 minutes can explain the problem.

2. Some minor grammar errors could be corrected during proofreading (also check for repetitions, see lines 175-176).

The relevant content in the article has been revised and highlighted in yellow.
